# Telomere Length, Apoptotic, and Inflammatory Genes: Novel Biomarkers of Gastrointestinal Tract Pathology and Meat Quality Traits in Chickens under Chronic Stress (*Gallus gallus domesticus*)

**DOI:** 10.3390/ani11113276

**Published:** 2021-11-16

**Authors:** Kazeem Ajasa Badmus, Zulkifli Idrus, Goh Yong Meng, Kamalludin Mamat-Hamidi

**Affiliations:** 1Department of Animal Science, Universiti Putra Malaysia, Seri Kembanga 43400, Selangor, Malaysia; alqasiminternational@yahoo.com (K.A.B.); zulidrus@upm.edu.my (Z.I.); 2Institute of Tropical Agriculture and Food Security, Universiti Putra Malaysia, Seri Kembanga 43400, Selangor, Malaysia; ymgoh@upm.edu.my; 3Department of Veterinary Pre-Clinical Science, Universiti Putra Malaysia, Seri Kembanga 43400, Selangor, Malaysia

**Keywords:** broilers, corticosterone, body weight, GI tract, meat qualities, biomarkers

## Abstract

**Simple Summary:**

The assessment of poultry’s gastrointestinal (GI) tract and meat quality traits are crucial for sustainable poultry production in the tropics. The search for well-conserved and more reliable biomarkers for the GI tract and meat traits has faced many challenges. In this study, we observed the effect of corticosterone (CORT) and age on body weight, buffy coat telomere length, GI tract, and meat quality traits. The critical evaluation of the GI tract and meat traits in this study revealed that telomere length, mitochondria, and acute phase protein genes were altered by chronic stress and were associated with the traits. This study informed us of the potential of telomere length, mitochondria, and acute phase protein genes in the assessment of GI tract pathological conditions and meat quality in the poultry sector for sustainable production.

**Abstract:**

This study was designed to examine the potentials of telomere length, mitochondria, and acute phase protein genes as novel biomarkers of gastrointestinal (GI) tract pathologies and meat quality traits. Chickens were fed a diet containing corticosterone (CORT) for 4 weeks and records on body weight, telomere length, GI tract and muscle histopathological test, meat quality traits, mitochondria, and acute phase protein genes were obtained at weeks 4 and 6 of age. The body weight of CORT-fed chickens was significantly suppressed (*p <* 0.05). CORT significantly altered the GI tract and meat quality traits. The interaction effect of CORT and age on body weight, duodenum and ileum crypt depth, pH, and meat color was significant (*p <* 0.05). CORT significantly (*p <* 0.05) shortened buffy coat telomere length. *UCP3* and *COX6A1* were diversely and significantly expressed in the muscle, liver, and heart of the CORT-fed chicken. Significant expression of *SAAL1* and *CRP* in the liver and hypothalamus of the CORT-fed chickens was observed at week 4 and 6. Therefore, telomere lengths, mitochondria, and acute phase protein genes could be used as novel biomarkers for GI tract pathologies and meat quality traits.

## 1. Introduction

Growth and body performance have been linked to skeletal muscle development. Intestine, liver, and muscle are metabolically active during growth and development [1]. Significant reduction in body weight of chronically stressed chicken has been reported [2]. Corticosterone has been used as a stressor agent to study the effect of stress on animal in various scientific disciplines. The reduced weight in corticosterone (CORT) treated chicken was reported to lead to enhanced gluconeogenesis and protein catabolism [3]. Findings have shown that CORT-treated chickens tend to divert resources from growth to lipid accumulation, hence their fatty livers [4]. A CORT diet could therefore be used as a proxy for animal stress to investigate biomarkers of meat quality traits and gastro intestinal tract physiology. In most studies, the small intestine is usually affected by chronic stress in chicken [5,6]. Reduction in the size of small intestinal epithelial was observed to contribute to decreased duodenal and jejunal villus height and crypt depth [5] in CORT-fed chickens. On the contrary, increased appetite in broiler CORT-fed chickens has been reported [6]. Increases in duodenal villi height were noticed in rats fed with glucocorticoid medication [7,8]. In addition, meat yield was reduced by CORT administration in chickens, and this led to subsequent suppressions in protein synthesis and protein catabolism in skeletal muscle [9]. Muscle fiber type, myofibril area, and muscle density affect meat biochemical compositions [10]. Lower lactate and glycogen levels might be indicative of oxidative activities, and this type of muscle might be characterized with a higher fiber concentration and smaller cross-sectional area of fiber (CSAF) [11]. Dysfunctional telomere length has been associated with protein damage [12], and effect of CORT has been reported to cause protein degradation [13]. The pH was linked to oxidative damage, and CORT administration has been reported to induce oxidative damage on DNA due to reactive oxygen species (ROS) that usually alter sugars and bases and cause DNA fusion [14,15]. In recent years, the concept of gut health (gastrointestinal tract) is attracting remarkable attention among nutritionists, veterinarians, and animal scientists to improve farm animals’ production performance [16]. The gastrointestinal (GI) tract comprises the luminal organs (esophagus, stomach, small intestinal and colon) and organs (pancreas, liver, gallbladder, and bile ducts), which connects and deliver to the luminal GI tract exocrine enzymes, bile salts, and acid to aid digestion [17]. The GI tract has been noted to be one of the organs that are mostly prone to cancer and tumors [17]. A panel of biomarkers to measure functionality and health of the GI tract in animals has been developed to solve the inherent challenges in choosing the right biomarker [18]. The search for highly conservative biomarkers that are associated with cancer and tumor in the GI tract and meat quality traits necessitated this study. We, therefore, proposed telomere length, mitochondria genes, and acute phase proteins as novel and well-conserved biomarkers for selecting meat with good qualities and assessing GI tract pathological conditions. Telomeres are nucleotides with repetitive sequences that capped the end of chromosomes, safeguarding them from degradation or fusion with other chromosomes [19]. Telomeres are usually shortened because of end-replication problems during cell division [20]. This shortening of telomeres is generally believed to be hastened by oxidative stress and inflammation [21]. Oxidative stress has been linked to inflammations and apoptosis, resulting in faster attrition of telomere length [22]. Telomere length has been measured in various red blood cells [23,24], lymphocytes [25,26], and whole blood [27,28]. Telomeres from various blood components have been related to aging, mortality, and ill-health conditions [21,23]. Moreover, understanding the expressions of mitochondrial DNAs is very important because of their roles in oxidative stress. Oxidative stress has been noted to be harmful to skeletal muscles. Higher numbers of reactive oxygen species (ROS) build-up in the mitochondria [29] and mitochondria uncoupling protein 3 (*UCP3*) are activated from numerous scenarios characterized by skeletal muscle deterioration, cancer, and glucocorticoid (GCs) administration [30]. Another component of mitochondrial DNA is Cytochrome C oxidase (*COX6A1*), which belongs to complex IV of the mitochondrial complexes. *COX6A1* is said to be necessary for apoptosis [31]. In addition, the acute phase protein, serum amyloid A 1 (*SAA1*), plays a crucial role in fat metabolism, phagocytosis, and in the regulation of inflammation and tumor effects. Another acute phase protein that has been associated with telomere length is C-reactive protein (CRP). The principal roles of CRP are that it recognizes and eliminates pathogens, clears cellular and nuclear wastes, binds apoptotic cells, and intensifies the classical pathway of complement activation [32]. Reports on how buffy coat telomere length, mitochondria genes, and acute phase proteins could predict the gastrointestinal tract pathological states and meat qualities are not available, thus necessitating this study. We, therefore, intended to test the hypothesis that changes in telomere length, mitochondria genes (*UCP3* and *COX6A1*), and acute phase protein genes (*SAAL1* and *CRP*) due to chronic stress alter GI tract pathological conditions and meat quality traits. Therefore, this study was designed to examine how buffy coat telomere length, *UCP3*, *COX6A1*, *SAAL1*, and *CRP* could be used as a novel biomarker of GI tract pathological conditions and meat quality traits.

## 2. Materials and Methods

### 2.1. Animal Management, Housing, and Experimental Design

This study was conducted in compliance with the Animal Utilization Protocol approved by the Institutional Animal Care and Use Committee (IACUC) of Universiti Putra Malaysia (Approval number: UPM/IACUC/AUP-R019/2018). A total of one-hundred (100)-day-old male Cobb500 chicks were used for this study and were subjected to a 2 × 2 completely randomized factorial design. Chicks were randomly assigned by treatment in groups of 10 into 10 battery cages and subsequently weighed and wing banded. All the cages were placed in a single environmentally controlled chamber. The cages area measured 122 cm in length, 91 cm in width, and 61 cm in height. The experiment began with an initial temperature fixed at 32 °C on day 1 and gradually lowered to 21 °C until day 21 and was maintained as such to the end of the experiment. Relative humidity ranged from 70% to 80%. The chickens were vaccinated with infectious bronchitis and New Castle disease virus vaccines at days 7 and 14, respectively.

### 2.2. Diets and Corticosterone Challenge

The diets of the experiment consisted of starter (crumble from) and finisher (pellet form), which were fed ad libitum. The diet contained crude protein (CP) of 21.0% in the starter phase and 19.0% in the finisher phase. Chickens were kept for the first 14 days without CORT diet (week 0) and then subjected to CORT feeding on day 15 of age (beginning of week 3) for another 28 days (4 weeks). The control group consisted of 50 chickens that were given a commercial diet free from CORT (Table 1). CORT (Abcam, Cambridge, UK) was used as an endogenous steroid hormone, which has the property of inducing apoptosis.

In this study, 20 mL ethanol was used to dissolve 500 mg CORT and thoroughly mixed with 16.67 kg of diet feedin a feed trough for each replicate. Mixing was frequently carried out prior to the consumption of the diets throughout the experiment. Both the treatment and the control diets were mixed with ethanol for elimination of bias. Only one supplemental level of CORT (30 mg/kg diet) according Hu and co-researchers was used [33]. Both the CORT treated group and the control comprised of five replicates with ten chickens each in a cage. The diets were administered for 4 weeks to ensure the effect of stress and monitor age effect.

### 2.3. Growth Rate Parameters and Animal Sampling

The body weights of all the chickens were measured weekly using a EB-300 3 Model—Dual Range weighing scale (CAS Corporation, Seoul, South Korea) (see Appendix A). At the end of week 2 and 4 of CORT administration, 2 chickens were sampled at random from each cage from the CORT and control groups for slaughtering, totaling 10 chickens per group. The slaughtering was humanely performed by severing the jugular veins, carotid arteries, trachea, and esophagus with a sharp knife by a single swipe. This sampling was carried out for telomeric DNA determination and histopathological evaluations at weeks 4 and 6 of age (end of the second and fourth weeks of CORT administration, respectively). The blood samples, due to exsanguination, were collected into EDTA tubes and immediately stored in ice. The blood samples were stored at −20 °C prior to DNA extraction. The intestinal samples (duodenum, ileum, and jejunum) were collected and stored in formalin (formaldehyde) for hematoxylin and eosin staining. The plasma, buffy coat, and erythrocyte were separated by centrifuging at 3000 rpm for 30 min at 25 °C. The buffy coat portions were scooped out of these three components using Eppendorf pipettes (Eppendorf, Hamburg, Germany) and then stored at −80 °C until used for DNA assay. Three portions of breast meat samples were collected. The first portion was stored on ice and later kept at −80 °C for meat quality traits. The second portion was stored in formalin for muscle fiber analysis, and the third portion was stored in liquid nitrogen and then transferred into −80 °C for gene expression analysis. In addition, for the gene expression, liver, heart, and brain (hypothalamus) were also collected into liquid nitrogen and transferred into −80 °C.

### 2.4. Histopathological Test of the Small Intestinal Villi Components, Muscle Myofibrils, and Liver

The small intestinal duodenum, jejunum, ileum, muscle, and liver samples of CORT-fed chickens and control chickens were stored in 10% formalin. All samples were processed, cut, embedded, trimmed, and sectioned with a microtome (LEICA Biosystems, Illinois, IL, USA). The samples were allowed to cool, then fixed on slides, incubated for 30 min, stained using hematoxylin (H) and eosin (E) methods, and subsequently washed with alcohols and xylenes of different concentrations (70–95%). The final stained samples were covered with a fine microscope slide and fixed using a mounting medium. The slide was then viewed using CMEX 1300 × ImageFocus v3.0.01 (Euromex microscope, Arnhem, Holland) at 4× magnification, and all measurements were made in micrometers. The scoring and grading system adopted by [34] was employed for the study of fatty liver. Single thicknesses of the fibers were considered to be myofibrils, and collections of it were measured as myofibril bundles (see Appendix A). The length of the villi from the bottom of the crypt depth to the top was measured as the villi height (see Appendix A). Villi crypt depth was measured from the base of the villus to the mucosa (see Appendix A). Small intestinal villi (duodenum, jejunum, and ileum), muscle, and liver samples were taken from all the sampled chickens from both the CORT-treated and control chickens. All the histopathological measurements were made directly using a CMEX 1300 × ImageFocus v3.0.01 camera (Euromex microscope, Arnhem, Holland).

### 2.5. Meat Quality Test

#### 2.5.1. Tissue Sample Collections

For meat quality traits and telomere length evaluation, breast muscle samples were collected from both the CORT-fed chickens and control chickens at weeks 2 and 4 of CORT administration, stored on ice, and transferred to a −80 °C freezer until analysis. For the gene expressions, breast muscle, liver, and brain samples were obtained and freeze-dried in liquid nitrogen and transferred immediately into a −80 °C freezer.

#### 2.5.2. Muscle pH Measurement

The pH values of breast muscles at 24 h post-slaughter were measured using a portable pH meter (AG 8603 Mettler Toledo, Schwerzenbach, Switzerland). A total of 1 gram of crushed sample was taken and homogenized for 10 s in 10 mL cold sodium iodoacetate dissolved in deionized water at maximum speed using a homogenizer (Wiggen Hauser D-500, Berlin, Germany). The pH of each sample was measured in triplicates, and the average reading of the replication was then determined (see Appendix A).

#### 2.5.3. Measurement of Drip Loss

Approximately 50 g of fresh breast muscle samples at weeks 2, 4, and 6 was taken, weighed with the initial weight (W1). The meat samples were kept in a plastic bag, vacuum packaged, and kept at 4 °C. The samples were removed from the bags after postmortem storage and gently blotted with tissue to dry, then weighed and recorded as W2. The percentage of drip loss was evaluated as: drip loss% = [(W1 − W2)/W1] × 100 [35] (see Appendix A).

#### 2.5.4. Measurement of Cooking Loss

The breast muscle samples were weighed and recorded as initial weight (W1). Then, the muscle samples were placed in a plastic bag, vacuum packaged, and cooked in a water bath at 80 °C for 20 min. The samples were then dried using tissue paper without pressing and weighed (W2). Cooking loss was measured as: cooking loss% = [W1 − W2/W1] × 100 (see Appendix A).

#### 2.5.5. Measurement of Meat Tenderness

Meat tenderness was evaluated from the sample of breast muscle used for the determination of cooking loss. The sample was stored in the refrigerator (4 °C) overnight in a plastic bag to prevent evaporation. On the next day, the cooked sample was cut into at least 3 subsamples (blocks) 1 cm high × 1 cm thickness × 2 cm length (±0.5 mm) with the long axis accurately aligned with the muscle fiber direction [36]. Each subsample was sheared perpendicular to the muscle fibers using a TA.HD plus^®^ texture analyser (Spectrometer technologies, Surrey, UK), fitted with a Volodkevitch blade set according to the procedure of [37]. Shear force values were documented in kilogram (Kg) as the average of all block values of each individual sample (see Appendix A).

#### 2.5.6. Measurement of Color

Breast muscle samples were bloomed (27 °C) for 25–30 min prior to the color analysis. The color coordinates were determined using ColorFlex spectrophotometer (Hunter Lab, Reston, VA, USA). The device was calibrated against black and white reference tiles prior to use. A total of three readings for each sample (the cup rotated 90 degrees in the second and third reading) of L * (lightness), a * (redness), and b * (yellowness) were recorded, and then the average value was calculated for each sample [38] (see Appendix A).

#### 2.5.7. Determination of Crude Protein

A portion of dried meat sample (1 g) was digested in sulfuric acid (H_2_SO_4_) with the addition of two Kjeldahl tablets which converted all the nitrogen (N) present into ammonia (NH_3_). The ammonia was obtained by adding sodium hydroxide to the digest, distilled off, and collected into standard acid with indicator. Hydrochloric acid was used for titration against the digest. The quantity collected was determined by automated titration with 6.25 factor using the FOSS Kjeltec analyser (2.0.0 version—Kjeltec^TM^ 8400) (FOSS, Hilleroed, Denmark) in percent crude protein (see Appendix A)

#### 2.5.8. Determination of Fat (Ether Extract)

A portion of meat sample (3 g) was subjected to continuous extraction with petroleum ether using the FOSS Soxtec apparatus (FOSS, Hilleroed, Denmark) for a 2 h period. The solvent was then evaporated, and the residue was then referred to as the resultant extract or crude fat using ether extract (see Appendix A).

#### 2.5.9. Determination of Energy

A part of the meat sample (1 g) was weighed and made into pellet. This pellet was then loaded onto the bomb calorimeter and then subjected to oxidation. The disappearance of the sample from the calorimeter indicated the total energy determination. The direct energy value was automatically generated in MJ/Kg (IKA C2000, Guangzhou, China) (see Appendix A).

#### 2.5.10. Moisture Contents and Dry Matter

The meat sample was cleaned, weighed (wet weight), wrapped in aluminum foil, and dried in the oven at 60 °C for 72 h. The subsequent weight of the dry sample was taken (dried weight) and subtracted from the wet weight. The percent of the difference divided by the original wet weight gave us the percent moisture content (MC%) (see Appendix A). 

Therefore, dry matter% = 100 − MC%

### 2.6. Determination of Telomere Length Using Real-Time Quantitative PCR (qPCR) Analysis

DNA was extracted from the separated buffy coat using blood DNA innuPREP Mini Kit (Analytik Jena, Jena, Germany) following the manufacturer’s recommendations. The qualities of the DNA extracted were tested using gel electrophoresis and nanodrop (Multiskan Go by Thermoscientific, Waltham, MA, USA). DNA samples with 1.8–2.0 (260/280 ratio) values were stored in a −20 °C freezer prior to the telomeric length determination analysis. For the discovery of the telomere length, the primers were adopted from an available report [39] (Table 2). The single copy gene (SCG), *GAPDH* (glyceraldehyde-3-phosphatase), primer [23] sequences were designed and sequenced using information contained in the gene bank (NCBI) specific to chicken (Table 2). The primers were tested with DNA amplified using MyTaqTM Red Mix (Bioline, London, UK) with the aid of the polymerase chain reaction (PTC-100TM, Marshall Scientific, Hampton, NH, USA) before running them for the qPCR analysis. Twenty nanograms of DNA template was used for both the telomere and the *GAPDH* reactions. The forward and reverse primer concentrations for both telomere and *GAPDH* were 2 µM of each. The primers were mixed with 10 µL SensiFAST SYBR No-ROX qPCR master mix (Bioline, London, UK) for a total volume of 20 µL. Ten-fold serial dilutions were performed to obtain standard curves for both the telomere and the housekeeping gene. The samples were arranged accordingly in the PCR machine with identifiers, including the non-template control (NTC). The cycling conditions for both telomere and the single-copy gene (SCG), *GAPDH* were: 10 min at 95 °C, followed by 40 cycles of 95 °C for 15 s and 60 °C for 1 min, followed by a dissociation (or melt) curve using CFX96 Real-Time PCR System (Bio-Rad, Hercules, CA, USA). Any cycle threshold (Ct) value of standard deviation above one was not used for these analyses. The amplification results (Ct values) were subjected to a Microsoft Excel program designed from standard curves generated to obtain copy numbers in kilobase per reaction (kb/reaction) of both the telomere and the SCG. The Kb/reaction values were then used to calculate the total telomere length in kb per chicken diploid genome according to available information [25] (see Appendix A).

### 2.7. Gene Expression Analysis of Mitochondrial DNA Related Genes and Acute Phase Protein Factors Using Reverse Transcriptase (RT)-PCR

RNA samples were extracted from the muscle, liver, heart, and hypothalamus according to the manufacturer’s protocols (InnuPREP RNA Mini kit 2.0, Analytik Jena AG, Jena, Germany). The qualities of the RNA were tested using nanodrop (Multiskan Go by Thermo Scientific, Waltham, Massachusetts, USA) and ribonucleic acid (RNA) with 1.8–2.0 (260/280) values were stored at −80 °C for further analysis. The conplementary DNA (cDNA) was synthesized using reverse transcription analysis from the extracted RNA (1µg) samples using the SensiFast cDNA synthesis kit according to the manufacturer’s protocol (Bioline, London, UK) on a PTC-100 PCR machine (MJ Research, Marshall Scientific, Hampton, NH, USA) and stored at −20 °C The cycling conditions for the reverse transcription were: 10 min at 25 °C for annealing, followed by 15 min at 45 °C of reverse transcription, 5 min at 85 °C of inactivation, and infinity at 5 °C. Gene expressions of the mitochondrial DNAs, uncoupling protein 3 (*UCP3*), and cytochrome c oxidase (*COX6A1*) were analyzed using quantitative PCR from the cDNA synthesized from the RNA extracted from the liver, muscle, and heart tissues. Genes controlling acute phase proteins, serum amyloid a (*SAAL1*) and C-reactive protein (*CRP*), were evaluated from the liver and hypothalamus tissues. The primers of these genes specific to chicken were designed using the data available in the gene bank of the National Centre for Bioinformatics (NCBI) (Table 2) and were optimized using a normal PCR and PCR master mix (MyTaqTM Red Mix, Bioline, London, UK) before the real-time PCR analysis. Both the test genes and the reference genes (*GAPDH* and *β-Actin*) were simultaneously run in a plate (96 wells) in duplicates. The concentrations of the primers used for the test and reference genes as determined by titration were 2 µM each for the forward and reverse primers. The samples were arranged accordingly in the PCR machine with identifiers including the NTC and the standards. The cycling conditions for both the test genes and the reference genes were as described for the telomere length determination above. The SensiFast Sybr^®^ No-ROX kit (Bioline, London, UK) master mix was used for the amplification of the DNA products. The Ct values with standard deviation above one were not used for these analyses. The mean fold change in the expression of a target gene at each time was calculated using the method described by the study in [40], with 2^-ΔΔCT^ where ΔΔCT = (CT, target test − CT, reference test) − (CT, target control − CT, reference control). The average of the two reference genes Ct values was taken using the geometric mean and then used to normalize the test genes (see Appendix A).

### 2.8. Statistical Analysis

The data on body weight, histopathology, meat quality traits, telomere lengths were subjected to 2 × 2 factorial analysis according to the procedure of the SAS 9.4 [41]. Mean separation among group was performed using the Duncan multiple range test. Gene expression data of *UCP3*, *COX6A1*, *SAAL1*, and *CRP* were analyzed using a box-plot test. The data were subjected to tests of normality, and they satisfied the conditions for normal distribution curves. The assessment of histogram distribution and quartile (Q) plots for the model residual was used for the assumption of normality. Log transformation was used for the gene expression data phenotypic correlation and regression analyses among telomere lengths, and meat quality traits were obtained by the analysis of covariance using the Pearson correlation procedure of SAS 9.4 [42]. The multiple hypothesis correction was conducted for all the variables using the Benjamini–Hochberg (BH) correction method (*p* < k/m × α).

## 3. Results

### 3.1. Body Weight and Buffy Coat Telomere Length

Interaction between treatment and age was very significant (*p <* 0.0001) (Table 3). There was no interaction effect of CORT and age on buffy coat telomere length. However, we noticed a significant (*p <* 0.05) effect of CORT and age on buffy coat telomere length. Significant effects of CORT on the body weight of CORT-fed chickens were noticed at both weeks 4 and 6. There were increments in body weight and telomere length in both the control and the CORT group along age lines.

### 3.2. Small Intestinal Duodenal, Jejunal, and Ileal Villi Height and Crypt Depth

There was no significant interaction effect between treatment and age on duodenum, jejunum, and ileum villi height and relative feed intake (RFI) (Table 4). However, a significant (*p* < 0.05) effect of CORT and age was revealed for duodenum villi height. A significant (*p* < 0.0001) effect of age was revealed for jejunum villi height, ileum villi height, and RFI. An effect of CORT was revealed for jejunum villi height (*p* < 0.05). A significant (*p* < 0.05) effect of CORT was only revealed for duodenum and jejunum villi height at week 4 but not at week 6. It was, however, revealed at both weeks 4 and 6 for RFI.

Significant (*p* < 0.05) interaction effects of treatment and age on duodenum and ileum crypt depth (Table 5) were noticed. A significant (*p* < 0.05) effect of age on duodenal crypt depth but not on jejunum and ileal crypt depth was observed. No significant effect of age and treatment on jejunum crypt depth was observed. A significant (*p* < 0.05) effect of CORT on duodenum and ileum crypt depth was observed at both weeks 4 and 6.

### 3.3. Muscle Myofibrils and Myofibril Bundle Diameters

In this study, no significant interaction effect of treatment and age on myofibrils and myofibril bundle diameters was observed (Table 6). However, we noticed a significant (*p* < 0.0001) impact of CORT and age on myofibrils. A significant (*p* < 0.001) effect of CORT on myofibril bundles was noticed in this study.

### 3.4. Liver Histopathology

The liver samples’ histopathological tests of CORT-fed chickens and control chickens are presented in Figure 1 and Table 7. The liver of the CORT group showed necrotic and tumor signs at both weeks 2 and 4 compared to the control. Severe non-alcoholic fatty liver (NAFL) cases were observed in the CORT-fed chickens compared to the control. The symptoms of the NAFL observed were fibrosis, ballooning, steatosis, and inflammation. None of the symptoms stated here were observed in the control chickens.

### 3.5. Meat Quality Traits

Significant (*p* < 0.0001) interaction effects of treatment and age on the degree of acidity and alkalinity (pH), lightness (L *), redness (a *), and yellowness (b *) were observed (Table 8). No interaction effects of age and treatment on drip loss (DL), cooking loss (CL), and shear force (SF) were observed between the CORT-treated chickens and the control. However, a significant effect of age and treatment on DL was observed. A significant (*p* < 0.05) effect of treatment was observed for SF. In addition, a significant (*p* < 0.05) effect of treatment on pH was noticed at both weeks 4 and 6. Effects of CORT were only observed at week 5 on L * and a * and at week 4 on b *.

No interaction effect of age and treatment on crude protein (CP), energy, ether extract (EE), moisture content (MC), and dry matter (DM) was noticed (Table 9). However, a significant (*p* < 0.05) effect of age on energy and EE was revealed, while a significant (*p* < 0.05) effect of treatment on MC and DM was revealed. No effect of treatment was noticed for CP, energy, and EE.

The results of the correlations between buffy coat telomere length and meat quality traits are presented in Table 10. A positive and significant correlation was observed between telomere length and meat pH (r = 0.60, *p* < 0.05) at week 2 of CORT administration. A negative and significant correlation was observed between telomere length and drip loss (r = −0.59, *p* < 0.05) at the same week. A correlation between drip loss and pH was negative and significant (r = −0.642, *p* < 0.05). A negative and significant correlation was reported between cooking loss and drip loss. Positive and significant correlations were observed between lightness and shear force (r = 0.589, *p* < 0.05) and yellowness and lightness (r = 0.587, *p* < 0.05). The prediction model equation of telomere length with pH and DL at week 4 are presented in Table 11. Linear regression analysis using adjustment R was used to predict the link between telomere length with both pH and drip loss. It revealed a significant prediction model for meat quality using telomere length. The pH at week 4 was significantly (*p* < 0.05) fitted to the regression equation line.

### 3.6. Mitochondrial Genes

#### 3.6.1. Uncoupling Protein 3 (UCP3)

The mitochondrial *UCP3* was significantly (*p* < 0.05) downregulated with 0.58 fold change (FC) and significantly (*p* < 0.05) upregulated (2.36 FC) in the muscle of the CORT-fed chickens at weeks 4 and 6, respectively (Figure 2). In addition, *UCP3* in the liver was upregulated (3.40 FC, *p* < 0.05) at week 4 but no change (*p* > 0.05) in expression was noticed at week 6. Upregulation (4.30 FC, *p* < 0.05) of *UCP3* was observed in the heart tissue at weeks 4, but no significant change in its expression was noticed at week 6.

#### 3.6.2. Cytochrome C Oxidase (*COX6A1*)

The cytochrome C oxidase (*COX6A1*) genes were upregulated in the liver (4.22 fold change (FC), *p* < 0.05) and heart (2.34 FC, *p* < 0.05) of the CORT-fed chickens at week 4 and 6, respectively (Figure 3). It was, however, downregulated (0.31 FC, *p* < 0.05) in the muscle at week 6, but no change in its expression was observed at week 4. No changes in expression for *COX6A1* were observed in the muscle at week 4 and liver and heart at week 6.

### 3.7. Acute phase Protein

#### 3.7.1. Serum Amyloid A like 1 (*SAAL1*)

The serum amyloid A like 1 (*SAAL1*) was significantly upregulated in the liver of the CORT-fed chickens at weeks 4 (1.30 FC, *p* < 0.05) and 6 (1.37 FC, *p* < 0.05) (Figure 4). In addition, *SAAL1* in the hypothalamus was upregulated (1.29 FC, *p* < 0.05) in the CORT-fed chickens at week 6 but was downregulated (0.61 FC, *p* < 0.05) at week 4.

#### 3.7.2. C-reactive Protein (*CRP*)

Our results revealed that C-reactive protein (*CRP*) was upregulated in liver of the CORT-fed chickens at weeks 4 and 6, but without significant differences (Figure 5). However, *CRP* was significantly upregulated (1.73 fold change, *p* < 0.05) in the hypothalamus at week 4 but did not show any change in hypothalamic expression at week 6 in the CORT-fed chickens.

## 4. Discussion

### 4.1. Body Weight and Telomere Length

In this study, the corticosterone (CORT)-treated chickens exhibited substantially suppressed body weight at both weeks compared to the control group. The suppression in the body weight could result from diverting nutrients to fat in CORT-fed chickens [2]. In addition, there was a significant telomere length shortening following CORT treatment. This is in accordance with the recent developments indicating oxidative stress due to CORT-triggered telomere length attrition in birds [21,23]. The lower telomere length obtained in this study at week 4 compared to week 6 could be attributed to increased metabolic activities at an early age (week 4), which brought about higher resting energy expenditure (REE), leading to attrition of telomeric DNA [43]. The interaction between metabolic activities and chronic stress induced by the CORT diet could also be attributed to shorter telomere length. The REE has been reported to be higher at an early age in metabolically active organs [44,45]. The CORT diet mimics real-life stressors and telomere length as biomarkers of stress could be sensitive enough for the real-life stressors. Higher mortality and disease rates related to stress are usually linked to telomere length attrition. Improvement in the telomere length with age obtained in the CORT-fed group in this study could be attributed to the immune system of the buffy coat and reduced metabolic activities. The buffy coat is a component of the blood that comprises leucocytes (white blood cells) and blood platelets. The leucocyte protects the animal’s system from foreign bodies, while the platelets provide blood-clotting components to injuries. Dexamethasone (a corticosterone medication) has been reported to increase the blood concentration of leucocytes together with a decrease in the concentration of monocytes [46]. Corticosterone administration might lead to increased buffy coat leucocyte concentration in the CORT-fed chickens, and this could proffer immune advantages to the body system and, hence, improve the telomere length.

### 4.2. Intestinal Histopathology

The small intestinal villi are vital in the digestion of nutrients in the body. It aids in the absorption of digested nutrients into the blood for proper assimilation. The interruption of the villi could impair the physiological function of the small intestine and cause impairment of growth. There are no reports on how telomere length is specifically related to intestinal morphology. In the current study, histological morphology of the small intestine indicated that villi height and crypt depth were affected by oxidative stress induced by corticosterone, and this was evident in inflammation observed in the small intestine. The increase in the duodenal and jejunal villi height in the CORT-fed chickens might be the reason for the increased appetite leading to their higher feed intake. Increased appetite has been reported in broiler chickens fed with CORT [4]. The small intestine aids in growth and animal performance [47]. The small intestine stands to be metabolically active [1], and its development can be affected by several elements, especially stress [48,49,50]. It was inferred that the reduced size of the small intestinal diameter due to CORT treatment contributed to lower duodenal, jejunal villus height and crypt depth [5]. However, this is contrary to an increased appetite earlier reported in broiler chickens fed with CORT [6]. Therefore, the increased relative feed intake could be attributed to higher villi height and hence increased appetite. However, the feed consumed could have been diverted to fat deposition instead of improved digestibility and hence weight loss [4]. Villus height is positively associated with villus surface area [51], and this implies that a reduction in villus height will subsequently reduce its absorption capacity [52]. A group of researchers revealed that CORT administration improved glucose and calcium absorption [53], and it was claimed that this might not be the reason for the reduced small intestinal surface area [5]. Increased absorption of nutrients can only be assumed by increased villi height.

The variation in our report compared to that obtained in study [5] could be attributed to breed difference (Arbor Acres randomized population versus Cobb 500 broiler chickens). In addition, observations available on small intestinal villi were made based on absolute small intestinal data [3,30], whereas going by the relative data, it will be observed that the CORT-fed chickens had higher feed consumption (relative feed intake and higher appetite than the control). Increased feed intake and appetite could be attributed to increased villi height as a result of inflammation in the GI tract due to CORT. We noticed that the increased size of the villi height and crypt depth were abnormal during our investigation. Our observations agreed with a finding in [7], in which the authors reported a great increase in duodenal villi height in rats fed with an oral administration of prednisone (glucocorticoid medication). Glucocorticoid administration in rats had to precocious changes in the small intestine, including increased intestinal weight, villus, and crypt height [8]. This could result from enhanced cell proliferation leading to mucosa hyperplasia, which occurs throughout the entire small intestine [8]. Our results, therefore, revealed inflammatory enlargement of the small intestinal villi height and duodenal and ileal crypt depth. Reduced duodenal and ileal crypt depth at week 6 in the CORT-fed chickens could be attributed to apoptosis. The CORT chickens with short telomeres revealed small intestinal villi height and crypt depth inflammation and proliferation. This implies that small intestinal villi are well associated with telomere length. In the current study, CORT with the apoptotic-inducing property as described by the manufacturer was employed. This means that injuries were caused to the CORT-fed chicken tissues, consequently resulting in necrotic swelling and enlarged sizes of the small intestine, its villi height, and crypt depth. This situation might lead to cloudy swelling and cell bursting due to vacuoles and nuclear damages [54]. This nuclear damage could be responsible for the telomere shortening as obtained in the CORT-fed chickens. Research on telomere length intestinal adenocarcinoma (ACA) is limited. However, upregulation of telomerase activities in 75% of tested periampullary cancer samples has been reported, including 100% duodenal carcinoma tissue samples with reports of higher telomerase activities in high tumor cells than small tumor cells [55]. Both shorter and long telomere length has been linked with an increased risk of GI tract cancer or tumor [17]. Unveiling the mitochondrial and acute phase protein gene factors mediating the proliferation in the intestines could bring about reasonable insights.

### 4.3. Liver Histopathology

In the current study, CORT-fed chickens with shorter telomere lengths revealed symptoms of ill health, especially those related to non-alcoholic fatty liver (NAFL) and cancer. Similar reports can be found in the literature [56,57,58]. Acquired risk factors (non-alcoholic steatohepatitis, cirrhosis, and hepatocellular carcinoma) have been noted to be responsible for telomere loss [59]. The results of the liver histopathological test obtained in this study showed that the livers of the CORT-fed chickens were affected with inflammation, NAFL, and apoptosis due to CORT elevation. The NAFL could have resulted from the precocious enlargement of the livers of the CORT-fed chickens reported in our previous study [43]. Enlargement of the liver in the CORT-fed chickens has been attributed to the accumulation of fat [4]. These pathological symptoms have been, in one way or the other, correlated with telomere length shortenings. It has been shown that cells undergoing apoptosis upon DNA damage exhibited rapid and dramatic telomeric sequence losses [60]. Telomere shortening has long been hypothesized to be a biomarker of aging and a potential mechanism behind carcinogenesis [61,62]. Telomere length shortening measured in periphery blood cells (PBL) has been associated with an increased risk of cancers of the head and neck and urinary bladder [63,64], renal [65], liver [66] and esophageal [67] cancers.

### 4.4. Muscle Histopathology

The growth of animals is reflected in the sizes of the muscle and the bones. An increase in the sizes of these components usually leads to an increase in the body size of the animals. Myofibrils are the major proteins of the muscle and are made up of approximately 80% water (water-soluble proteins, free water, and bound water). The loss in telomere has not been associated with myofibrillar muscle damage, but protein damage in dysfunctional telomeres has been reported [12]. The results of the current study revealed that CORT caused a reduction in myofibrils and myofibril bundle diameters. The effect of corticosterone was investigated on myofibril protein of rats, and a 68% and 95% myofibrillar protein breakdown rate in both the non-diabetic and diabetic rats was observed, respectively [13]. The loss in the myofibrillar diameter could be attributed to high exudative loss (drip loss) accompanied by a loss in water-soluble proteins and accelerated muscle protein breakdown [68]. In this study, we noticed that muscle myofibrils increased with age, implying that damage caused by CORT was more devastating at week 4 (2 weeks of CORT treatment). It has been reported that that loss in the muscle myofibrils contributed to the loss in weight in the chronically stressed birds [68]. The loss in the myofibrils is associated with aging and aging-related diseases and could lead to apoptosis. This implies that telomere attrition could be experienced in the tissues of the CORT-fed chickens. We found out in this study that myofibrils improved with advancement in age. It was earlier reported in this study that telomeres improved in length as animals improved in age, and this could suggest a link between telomere length and muscle myofibrils. Low myofibril diameter indicates the lower cross-sectional area of fiber in the CORT-fed chickens, and this might imply that the CORT-fed chicken muscles are highly oxidative [11,69].

### 4.5. Meat Physical Components

The results of the meat quality traits revealed that corticosterone (CORT) administration affected the meat quality parameters assessed. The report in this study revealed that CORT was more devastating to meat at an early stage. This is evident as meat of the CORT-fed chickens tended to be more acidic at week 4. Acid muscles have been described as those with low ultimate pH (pHu < 5.7) [70], and this could trigger structural alterations in the muscles (apoptosis) of the CORT-fed chickens. Decreased pH and water holding capacity with increased proteolysis in broiler chickens injected with corticosterone was reported [71]. The degree and the level of the pH could influence both the taste and the structural features of the meat. Reduced ROS levels and increased telomerase activities and telomere length have been observed in mice blood whose diet was supplemented with alkaline water [72]. CORT-fed chickens also showed damaged meat as the pH was above the normal ultimate pH value (pHu > 5.8).

The L * and a * were both altered by the corticosterone in the CORT-fed chickens. These chickens revealed higher redness (a *) in their meat than the control and reduced L * at week 6. Redness (darkness) is a characteristic of poor meat and could be attributed to higher pH above the ultimate pH (5.8) as obtained at week 6 in this study. This type of meat could reveal firmness, water loss, and sensitivity to microbial degradation [73]. The darkness observed in the CORT-fed chickens could be the result of activated myoglobin induced by corticosterone, which was responsible for the dark coloration and susceptibility to oxidative metabolism [74]. This type of meat has a short shelf life, poor flavor, and is considered wasted in most cases by butchers. The CORT administration also increased the drip loss of the CORT bird meat at both weeks 4 and 6, which could be an indication of low water-holding capacity (WHC) of the meat, which is a typical sign of poor meat quality. As water is lost from the myofibrillar (<80% water) component of the muscle many substances of high quality are lost, such as water soluble proteins, free water, bound water, and entrapped water. Lower cooking loss and increased shear force were noticed in the chickens under transportation stress as compared to the control [75]. The results of our study agree with a report that observed a significantly increased shear force in long-term transported turkeys [76]. The meat obtained in the study revealed low tenderness as CORT contributed to higher shear force in the CORT-fed chickens at week 6.

### 4.6. Meat Biochemical Components

The biochemical profile data of the meat revealed that CORT did not have any effect on the meat proximate composition of broilers. The crude protein, lipid, and energy values were not affected by CORT administration. These results are different from those obtained whereby effects of sex on chicken meat quality were reported [77]. A study outlined the influence of the breed and age of chickens on the lipid content of chicken breast meat [78]. Our study indicated that corticosterone did not change the nitrogen, lipid, and metabolizable energy content of the meat. Increased protein levels and lowered lipid and ash levels in chicken of Thai origin have been documented [79]. Our experiments revealed that CORT can reduce the growth performance of birds and physical composition of the meat but not the biochemical composition. Increases in energy and ether could be attributed to increases in fat deposition in the circulation due to CORT. CORT had been implicated in the diversion of nutrients to fat deposition in broiler chickens [2].

### 4.7. Relationship between Telomere Length and Meat Quality Traits

The results of the correlations among telomere length, pH, and drip loss that we obtained revealed that corticosterone had pleiotropic effects, and there was an association between telomere length and meat quality traits. Therefore, telomere length could be used as a biomarker for monitoring meat quality traits and for selecting animals with good-quality meat. Correlations among meat quality traits revealed that a strong link exists between meat quality traits. The prediction model equation used in this study indicated that telomere length could be predicted by the pH of the meat. It was reported that a low amount of telomerase attributed to low extracellular pH was associated with short telomeres, which led to accelerated cell death and senescence [80,81].

### 4.8. Mitochondria Genes

Uncoupling protein (*UCP3*) and cytochrome C oxidase (*COX6A1*) are the mitochondrial genes that promote proton and electron transports in oxidative phosphorylation. In most cases, *UCP3* regulates cellular metabolism and oxidative stress. Uncoupling protein’s (*UCP3*) association with apoptotic, diabetic, and fatty acid cells has been investigated [82], but its role in skeletal muscles and the gastrointestinal tract (GIT) has not been elucidated. However, its role in apoptosis is implicated in meat quality. Our findings in this study indicate that the CORT-fed chickens presented acidic meat. It has been reported that acidic medium leads to tumor progression and apoptosis [83]. In addition, *UCP3* is required for mitochondrial maintenance. However, under severe conditions, it can lead to an increase in ROS, which is an agent of oxidative stress and DNA damage. Our results reveal diverse expression of *UCP3* and *COX6A1.* It was noticed that *UCP3* and *COX6A1* were upregulated in the liver and heart of CORT-fed chickens, and this could be the reason for the fatty liver symptoms in the liver. The expression of *UCP3* could be responsible for the poor meat quality traits in the CORT-fed chickens. The duo, especially *COX6A1*, is found in the pathway of apoptosis [84]. The downregulation of *COX6A1* in muscles at week 6 could imply mitochondria DNA dysfunction. The administration of CORT induced oxidative stress, which activated reactive oxygen species (ROS), and this could be attributed to the upregulation of *UCP3*. It is known that most ROS production in whole cells takes place in the mitochondria due to overexpression of *UCP3* [85]. Corticosterone administration has been associated with increased abdominal fat, and this could be enough to give rise to ROS, which causes injury to myocardial structure and physiology [86,87]. Studies have revealed that upregulation of *UCP2* and *UCP3* due to increased elevation of free fatty acids and mitochondrial ROS generation, as well as in the case of advanced obesity, caused cell death and heart seizure [88]. Moreover, several studies have examined yeast transfected with *UCP3* and discovered reduced growth rates and mitochondrial capacity, both of which suggested uncoupling activity [89,90,91]. Spontaneous ROS (superoxide, hydrogen peroxide, and hydroxyl peroxidase) generation has been reported to cause oxidative damage to DNA and protein [92]. In addition, both biological and genetic findings support the fact that *COX6A1* enhances apoptosis. Purified cytochrome C oxidase can induce caspase action in a cell-free system using extracts from healthy cells free from apoptosis [93]. Our results showed that *COX6A1* was upregulated, and this might have paved the way for apoptotic damage in the liver meat and other tissues, if well investigated, due to its apoptotic influences. Therefore, expressions of *UCP3* and *COX6A1* as functions of mitochondrial metabolic cellular activities could be the mechanisms behind the poor performance, shorter telomere length, inferior meat qualities, apoptosis, and inflammations of the GI tract of the CORT-fed chickens obtained in this study.

### 4.9. Acute Phase Proteins

Another important gene usually associated with telomere length is serum amyloid A like 1 (*SAAL1*). It is usually found in abundance in cells suffering from inflammatory injury. The results of the *SAAL1* and C-reactive protein (*CRP*) expressions revealed that both genes were activated in the liver and the hypothalamus of CORT-fed chickens, and these results indicated that they suffered from inflammation due to oxidative insults. It was explained in an early study that activation of serum amyloid A like 1 (*SAAL1*) mainly affected the acute phase response, a response due to environmental insults such as tissue damage, infection, and surgery [94]. Expressions of *SAAL1* in both the liver and the brain of the CORT-fed chickens indicated some level of injury in these tissues caused by reactive oxygen species (ROS), which led to the short telomeres and inflammation of the meat and the gastrointestinal tract found during the acute phase response. It has been noted that the upsurge in serum levels of serum amyloid A is triggered by physical injury to the host, including infection, trauma, inflammation, and cancer [95]. The upregulation of *SAAL1* in the CORT-fed chickens was expected due to the dark coloration of the meat of the CORT-fed chickens, a symptom of inflammation, and this is needed to provide immune cells to the areas of inflammation. In addition, *CRP* expressions in cells or tissues suffering from severe inflammation has been documented [96]. The upregulation of *CRP* in the tissues of the CORT-fed chickens indicated that corticosterone administration caused oxidative damage in these tissues. *CRP* is usually activated at the time of inflammation and infection [96]. Expression of *CRP* in the hypothalamus is a manifestation of oxidative damage in both the liver and the hypothalamus at both weeks 4 and 6 of CORT administration. The downregulation of *CRP* in the hypothalamus at week 4 might be a result of an immune system peculiar to the hypothalamus, and *CRP* could be highly expressed in the plasma and the serum [97,98]. In states of inflammation and oxidative stress, cellular damage is usually increased and could lead to accelerated telomere erosion [22]. The same authors discovered that the increase in plasma *CRP* concentration was associated with a decrease in telomere length. Upregulation of *SAAL1* and *CRP* in the broiler chickens in this study resulted in poor meat quality. Both genes are signs of inflammation, which is a typical symptom of damaged tissues.

## 5. Conclusions

Corticosterone suppressed body weight and shortened telomere length. It also caused oxidative damage to the meat traits, inflammation in the small intestine, and non-alcoholic fatty liver (NAFL) in the liver sample. Shortening of telomere length was associated with GI tract pathological symptoms and poor meat quality traits. Genes responsible for apoptosis (*UCP3* and *COX6A1*) and inflammation (*SAAL1* and *CRP*) were both expressed in the liver (GI tract) with NAFL and poor meat quality. Therefore, it is concluded that corticosterone affects the GI tract and meat quality traits, and the effects altered telomere length and expressions of *UCP3*, *COX6A1*, *SAAL1*, and *CRP.* Hence, telomere length and apoptotic and inflammatory genes could be used as novel biomarkers for the assessment of GI tract pathological conditions and meat quality traits.

## Figures and Tables

**Figure 1 animals-11-03276-f001:**
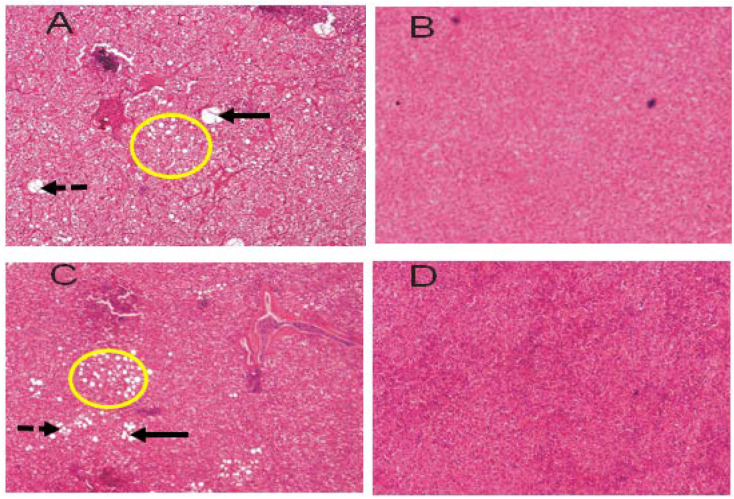
Liver test of non-alcoholic fatty liver (NAFL) and the control. (**A**): Liver of a CORT-fed chicken at week 2 of CORT administration; (**B**): liver of a control chicken at week 2 of CORT administration; (**C**): liver of a CORT-fed chicken at week 4 of CORT administration, and (**D**): liver of a control chicken at week 4 of CORT administration. Liver samples of the chickens exposed to corticosterone (CORT) feeding showed pathological symptoms. (**A**,**C**) show pathological symptoms (fibrosis, ballooning, inflammation, and steatosis) while (**B**,**D**) do not. Exposure of chickens to 30 mg/kg CORT diet caused apoptosis and non-alcoholic fatty liver. H&E, 4x. Dark arrows = fibrosis and steatosis; dotted arrows = ballooning; circle = inflammation.

**Figure 2 animals-11-03276-f002:**
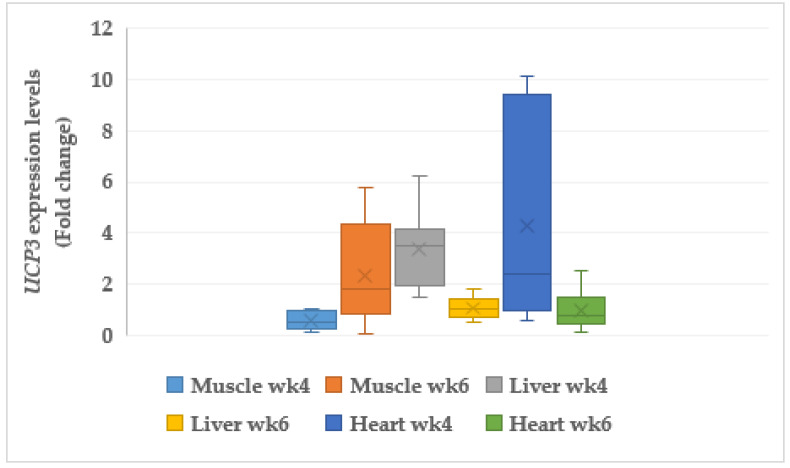
Expression profiles of mitochondrial uncoupling protein 3 (*UCP3*) in the muscle, liver, and heart at week 4 and 6 in corticosterone-fed chickens (fold-change). wk4 = 4 weeks of age (2 weeks of CORT administration); wk6 = 6 weeks of age (4 weeks of CORT administration); *n* = 20.

**Figure 3 animals-11-03276-f003:**
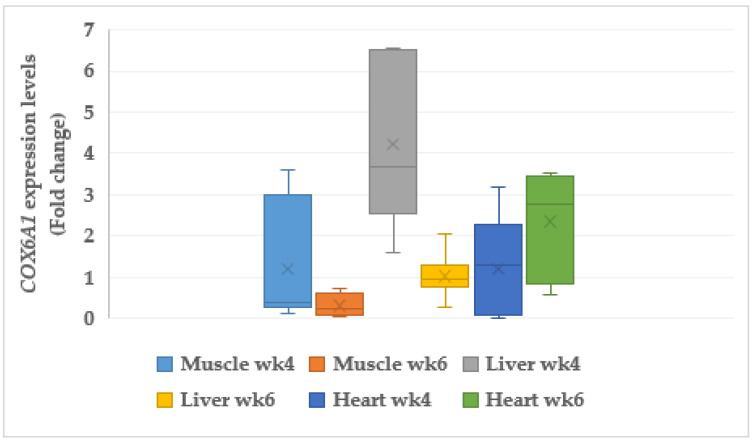
Expression profiles of mitochondrial cytochrome C oxidase (*COX6A1*) in the muscle, liver, and heart at week 4 and 6 in corticosterone-fed chickens (fold-change). wk4 = 4 weeks of age (2 weeks of CORT administration; wk6 = 6 weeks of age (4 weeks of CORT administration); *n* = 20.

**Figure 4 animals-11-03276-f004:**
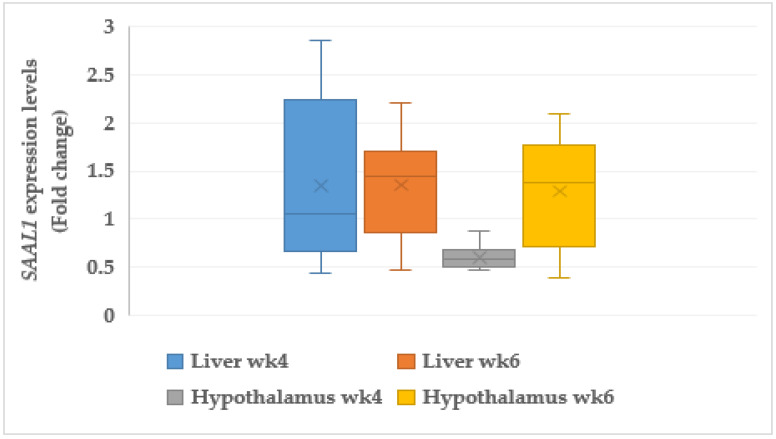
Expression profile of serum amyloid A like 1 (*SAAL1*) in the muscle, liver, and heart at week 4 and 6 in corticosterone-fed chickens (fold-change). wk4 = 4 weeks of age (2 weeks of CORT administration); wk6 = 6 weeks of age (4 weeks of CORT administration); *n* = 20.

**Figure 5 animals-11-03276-f005:**
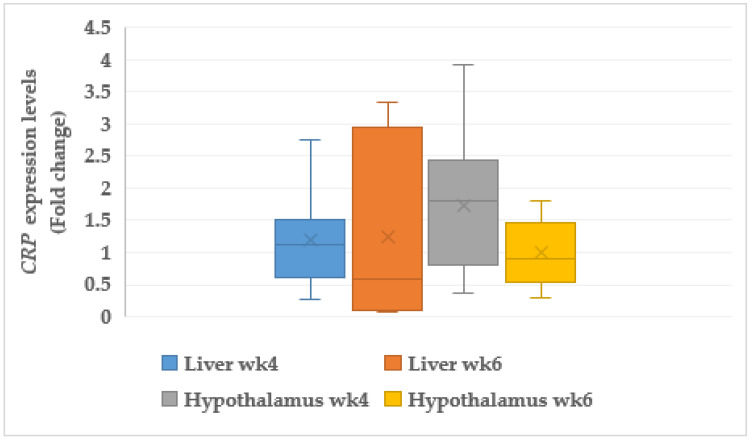
Expression profile of C-reactive protein (*CRP*) in the muscle, liver, and heart at week 4 and 6 in corticosterone fed chickens (fold-change). wk4 = 4 weeks of age (2 weeks of CORT administration; wk6 = 6 weeks of age (4 weeks of CORT administration); *n* = 20.

**Table 1 animals-11-03276-t001:** Nutrient compositions of commercial broiler starter and finisher diets.

Composition (%)	Starter ^1^	Finisher ^2^
Crude protein	23.00	19.00
Crude fiber	5.00	5.00
Crude fat	5.00	5.00
Moisture	13.00	13.00
Ash	8.00	8.00
Calcium	0.80	0.80
Phosphorous	0.40	0.40

^1^ Commercial starter diet for broiler in crumble form; ^2^ commercial finisher diet for broiler in pellet form.

**Table 2 animals-11-03276-t002:** Primer sequence for telomere regulatory genes in chicken.

S/N	Genes	Primer Sequence	Accession Number	Size (bp)
1	Telomere	**F**-GGTTTTT-GAGGGTGAGGGTGAGGGTGAGGGTGAGGGT **R**-TCCCGACTATCCCTATCCCTATCCCTATCCCTATCC-CTA	NA	79
2	*COX6A1*	**F**-TTCGACTGGGAGGACCATAG **R**-TGTTTTTCTGGGACACACCA	NC_-_006102.5	166
3	*UCP3*	**F**-AAGGATGGAGGTGTCCACAG **R**-GTGAGGAATACCCGGACTCA	NC_-_006088.5	161
4	*SAAL1*	**F**-GCCACCTCTCAAACTCTTGC **R**-CCTGCTTGTTTCCAGGAGAG	NC_-_006092.5	240
5	*CRP*	**F**-CCTAGGAGAACTGGGGAAGG **R**-CGTAGGAGAAGAGGCTGTGG	NC_-_006112.4	206
6	*GAPDH*	**F**-ACTATGCGGTTCCCAGTGTC- **R**-TGCCACCATCAGAAAAATGA	NC_-_006088.5	215
7	*β-Actin*	**F**-ACTGGATTTCGAGCAGGAGA **R**-CAGTGGAATGGGACAGACCT	NC_-_006101.5	248

NA = not available.

**Table 3 animals-11-03276-t003:** Effect of corticosterone administration and age on body weight and buffy coat telomere length (±SD) of broiler chickens.

Age	Treatment	Body Weight (g)	Telomere Length (Kb/Diploid Genome)
Week 4	CTRL	1509.36 ^b^ ± 49.15	248.23 ^ab^ ± 155.67
	CORT	1054.21 ^c^ ± 55.82	214.74 ^b^ ± 111.17
Week 6	CTRL	2363.55 ^a^ ± 100.11	466.36 ^a^ ± 262.56
	CORT	1479.18 ^b^ ± 354.72	235.44 ^ab^ ± 81.37
*p* values	Age	<0.0001	0.035
	Treatment	<0.0001	0.020
	Age × Treatment	<0.0001	0.78

^a^, ^b^, ^c^ Means within a column subgroup with no common superscripts are significantly different at *p* < 0.05; SD = Standard deviation. Note: CTRL = control; CORT = corticosterone. Week 4 and 6 of age are equal to 2 and 4 weeks duration of CORT administration to the chicken.

**Table 4 animals-11-03276-t004:** Effect of corticosterone administration and age on duodenum, jejunum, and ileum villi heights (±SD) of broiler chickens.

Age	Treatment	Duodenum Villi Height (µm)	Jejunum Villi Height (µm)	Ileum Villi Height (µm)	Relative Feed Intake
Week 4	CTRL	508.78 ^b^± 118.67	380.95 ^b^ ± 81.72	302.55 ^b^ ± 77.20	65.17 ^b^ ± 1.29
	CORT	633.16 ^a^ ± 134.95	441.96 ^ab^ ± 125.06	319.27 ^b^ ± 115.40	80.11 ^a^ ± 3.81
Week 6	CTRL	457.99 ^b^ ± 105.38	578.36 ^a^ ± 93.50	401.54 ^a^ ± 84.16	44.70 ^d^ ± 1.17
	CORT	496.00 ^b^ ± 155.49	608.22 ^a^ ± 130.71	388.00 ^a^ ± 137.95	55.59 ^c^ ± 2.14
*p* value	Age	0.033	<0.0001	0.0001	<0.0001
	Treatment	0.033	0.050	0.938	<0.0001
	Age × Treatment	0.195	0.555	0.474	0.103

^a^, ^b^, ^c^ Means within a column subgroup with no common superscripts are significantly different at *p* < 0.05; SD = Standard deviation. Note: CTRL = control; CORT = corticosterone. Week 4 and 6 of age are equal to 2 and 4 weeks duration of CORT administration to the chicken.

**Table 5 animals-11-03276-t005:** Effect of corticosterone administration and age on duodenum, jejunum, and ileum villi crypt depth (±SD) of broiler chickens.

Age	Treatment	Duodenum Villi CD (µm)	Jejunum Villi CD (µm)	Ileum Villi CD (µm)
Week 4	CTRL	68.14 ^ab^ ± 18.18	68.59 ± 15.49	50.83 ^ab^ ± 13.08
	CORT	80.17 ^a^ ± 18.40	59.38 ± 13.60	61.20 ^a^ ± 23.17
Week 6	CTRL	67.23 ^ab^ ± 23.60	57.67 ± 7.72	56.93 ^ab^ ± 9.63
	CORT	54.75 ^b^ ± 11.56	62.80 ± 20.43	48.46 ^b^ ± 11.79
*p* value	Age	0.025	0.564	0.359
	Treatment	0.475	0.886	0.752
	Age × Treatment	0.029	0.129	0.021

^a^, ^b^ Means within a column subgroup with no common superscripts are significantly different at *p* < 0.05; SD = Standard deviation. CD: crypt depth. Note: CTRL = control; CORT = corticosterone. Week 4 and 6 of age are equal to 2 and 4 weeks duration of CORT administration to the chickens.

**Table 6 animals-11-03276-t006:** Effect of corticosterone administration and age on muscle myofibrils and myofibril bundle diameters (±SD) of broiler chickens.

Age	Treatment	Myofibril (µm)	Myofibril Bundle (µm)
Week 4	CTRL	74.29 ^a^ ± 10.88	568.20 ^a^ ± 272.29
	CORT	52.89 ^b^ ± 3.27	427.50 ^b^ ± 136.89
Week 6	CTRL	88.96 ^b^ ± 10.68	773.90 ^a^ ± 176.68
	CORT	79.61 ^ab^ ± 6.35	442.60 ^b^ ± 153.69
*p* value	Age	<0.0001	0.134
	Treatment	<0.0002	<0.003
	Age × Treatment	0.827	0.194

^a^, ^b^ Means within a column subgroup with no common superscripts are significantly different at *p* < 0.05; SD = Standard deviation. Note: CTRL = control; CORT = corticosterone. Week 4 and 6 of age are equal to 2 and 4 weeks duration of CORT administration to the chicken.

**Table 7 animals-11-03276-t007:** Scoring and grades of non-alcoholic fatty liver in the CORT-fed chickens versus control chickens.

Symptoms/Grades	CTRL	CORT
Week 2		
Fibrosis	0	2
Ballooning	0	2
Inflammation	0	2
Steatosis	0	3
NAFL	0	Severe
Week 4		
Fibrosis	0	2
Ballooning	0	2
Inflammation	0	2
Steatosis	0	3
NAFL	0	Severe

NAFL = non-alcoholic fatty liver. Note: CTRL = control; CORT = corticosterone. Week 4 and 6 of age are equal to 2 and 4 weeks durations of CORT administration to the chicken (*n* = 20).

**Table 8 animals-11-03276-t008:** Effect of corticosterone administration and age on physical components of CORT chicken meat (±SD) versus control.

Age	Treatment	pH	L *	a *	b *	DL%	CL%	SF (Kg)
Week 4	CTRL	5.70 ^b^ ± 0.12	53.84 ^a^ ± 3.07	7.39 ^a^ ± 1.11	21.93 ^b^ ± 1.00	1.95 ^c^ ± 0.47	30.83 ^a^ ± 9.97	1.56 ^ab^ ± 0.39
	CORT	5.46 ^a^ ± 0.09	54.18 ^a^ ± 2.21	7.60 ^a^ ± 1.27	23.82 ^a^ ± 1.67	3.11 ^b^ ± 0.81	25.47 ^a^ ± 3.57	1.61 ^ab^ ± 0.49
Week 6	CTRL	5.81 ^b^ ± 0.20	55.01 ^a^ ± 3.38	5.34 ^b^ ± 1.80	20.73 ^b^ ± 2.78	3.57 ^b^ ± 0.59	31.14 ^a^ ± 8.82	1.33 ^b^ ± 0.16
	CORT	6.13 ^a^ ± 0.26	50.22 ^b^ ± 3.91	6.95 ^a^ ± 2.06	20.73 ^b^ ± 2.92	5.16 ^a^ ± 1.53	29.93 ^a^ ± 8.21	1.75 ^a^ ± 0.48
*p* value	Age	0.0001	0.046	0.0001	0.0001	0.0001	0.514	0.674
	Treatment	0.345	0.0003	0.002	0.047	0.0001	0.110	0.043
	Age × Treatment	0.0001	0.0001	0.021	0.029	0.518	0.257	0.090

^a^, ^b^, ^c^ Means within a column subgroup with no common superscripts are significantly different at *p* < 0.05; SD = Standard deviation. Note: L * = lightness; a * = redness; b * = yellowness; DL = drip loss; CL = cooking loss; SF = shear force; Kg = Kilogram; CTRL = control; CORT = corticosterone. Week 4 and 6 of age are equal to 2 and 4 weeks durations of CORT administration to the chicken (*n* = 20).

**Table 9 animals-11-03276-t009:** Effect of corticosterone administration and age on biochemical compositions (±SD) of CORT chicken meat versus control.

Age	Treatment	CP	Energy	EE	MC	DM
Week 4	CTRL	19.37 ± 3.58	20.67 ^b^ ± 0.41	5.29 ^b^ ± 1.73	74.32 ^a^ ± 1.50	25.68 ^b^ ± 1.50
	CORT	18.04 ± 0.94	20.90 ^ab^ ± 0.52	6.81 ^ab^ ± 2.22	72.89 ^b^ ± 1.51	27.11 ^a^ ± 1.51
Week 6	CTRL	18.80 ± 0.92	21.48 ^a^ ± 0.24	8.57 ^a^ ± 3.15	74.14 ^a^ ± 0.52	25.86 ^b^ ± 0.52
	CORT	18.50 ± 0.63	21.15 ^ab^ ± 0.62	8.41 ^a^ ± 1.33	72.84 ^b^ ± 1.06	27.16 ^a^ ± 1.06
*p* value	Age	0.879	0.014	0.017	0.782	0.768
	Treatment	0.310	0.746	0.480	0.002	0.002
	Age × Treatment	0.468	0.197	0.389	0.878	0.878

^a^, ^b^ Means within a column subgroup with no common superscripts are significantly different at *p* < 0.05; SD = Standard deviation. Note: CP = crude protein; EE = ether extract; MC = moisture content; DM = dry matter; CTRL = control; CORT = corticosterone. Week 4 and 6 of age are equal to 2 and 4 weeks durations of CORT administration to the chicken (*n* = 20).

**Table 10 animals-11-03276-t010:** Correlations between telomere length and meat quality traits of the CORT chickens versus control chickens at week 2.

	Telo	pH	SF	DL	CL	L *	a * b *
Telo							
pH	0.548 *						
SF	−0.091	−0.259					
DL	−0.496 *	−0.642 *	−0.387				
CL	0.183	0.238	−0.026	−0.570 *			
L *	−0.196	−0.133	0.589	−0.127	0.132		
a *	−0.103	0.144	−0.460	−0.096	−0.092	−0.480	
b *	0.091	−0.347	0.250	0.056	−0.208	0.597	−0.014

Correlation coefficient using Pearson correlation, Telo = telomere length; SF = shear force; DL = drip loss; CL = cooking loss; L * = lightness; a * = redness; b * = yellowness; *n* = 20. * = *p* < 0.05.

**Table 11 animals-11-03276-t011:** The prediction model equations of telomere length with pH and DL.

	Prediction Model	*p* Value
Telomere and pH	TeloWK4 = 5.426 + 0.00042 pH	0.028
Telomere and DL	TeloWK4 = 3.392 − 0.0022 DL	0.051

TeloWK4 = telomere at week 4 of age (2 weeks of CORT treatment); pH = acidity and alkalinity level; DL = drip loss; *n* = 20.

## Data Availability

The data presented in this finding were made available upon request from the corresponding author.

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
