# Peer review of "Telomere Length, Apoptotic, and Inflammatory Genes: Novel Biomarkers of Gastrointestinal Tract Pathology and Meat Quality Traits in Chickens under Chronic Stress (Gallus gallus domesticus)"

_animals, 2021, doi:10.3390/ani11113276_

Round 1
Reviewer 1 Report
In the present paper, the authors investigated the effect of corticosterone (CORT) on telomere length and meat quality traits in chickens. This article may be of relevance to the field, providing alternative tests for meat quality. However, the authors should improve the statistical analysis, and should provide more insight into the data. In addition, the gene expression analysis is not state of the art and the results may therefore be biased due to the method of normalization used (single reference gene only).
Although it is not entirely clear from the introduction. I assume that the authors used the CORT diet as a proxy for animal stress, to investigate biomarkers for meat quality traits. If this is correct, the authors should state this more clearly in the introduction. In addition, if this is the case, the authors should indicate whether this CORT diet is equivalent to the stressors that will act on chickens in a real life situation. The authors report some effects of the CORT diet on Telomere length, suggesting telomere length as a biomarker, but it should be discussed whether this biomarker will be sensitive enough for real life stressors.
The treatment group received CORT that was diluted in 20ml of ethanol and mixed in the feed. Did the control group also receive 20ml of ethanol mixed in the feed? This may be essential, as ethanol may also have an effect on the liver and may thus bias the results of the treatment group.
I have some comments on the statistical design of the experiments. The authors used t-tests and checked the assumption of normality, but they do not mention how this was checked and neither do they report whether the residuals were also normally distributed and whether the variation was homoscedastic. Given the relative small sample size, I doubt that the assumptions will be met for all tests. As such, I would strongly suggest the authors to disclose all raw data and the results of the statistical tests in a supplemental file.
In addition, the authors should provide the exact p-values, instead of merely stating p<0.05, at least for all –values under between 0.001 and 0.05. Furthermore, the authors should perform multiple hypothesis correction. Given the evaluation of a large number of hypothesis on a relatively small number of samples, the risk of false positives is high.
The authors should not mention the standard error of the mean (SEM). Reporting SEM is considered as outdated as it disguises the real variation that is present in the samples. The actual standard deviation for each of the measures or the 95% confidence intervals, if non-parametric statistics would be considered should be provided.
I would also urge to make the data more visible. Tables with only the means harbour very little room for interpretation. A better representation would be given using box-plots with the actual data points plotted on top of them. That would give the readers a good idea of the real distribution and the effect of each of the tested hypotheses.
The telomere lengths seem to be higher in the week 6 versus week 4 results. This is counterintuitive, but could of course be due to variation in the measurement procedure, but this is not clear and not discussed. Giving the raw data and a visual representation of the data (boxplots) would enhance the understanding of this variation.
The authors should carefully check the MIQE guidelines (https://academic.oup.com/clinchem/article/55/4/611/5631762) for reporting the experimental procedures and data of the quantitative PCR experiments. These guidelines provide a checklist of the minimal information that should be given when using these methods.
I have serious doubts about the gene expression analysis results. The authors only used a single reference gene (GAPDH) to normalize their data. This is a very outdated method of performing RT-qPCR and may result in biased results as the authors an by no means assume that the reference gene itself is not subject to variation between the experimental procedures. Already since 2002, the RT-qPCR field urges researchers to pre-validate reference gene stability from an initial set of at least 6-10 reference genes and use at least the two most stable reference genes for normalization.
The graphs of he RT-qPCR results should be shown as boxplots instead of barcharts, as the latter only hide relevant information.
I am a bit surprised that standard t-tests could be used on the RT-qPCR data. These data are frequently logarithmical in nature, hence transformation is often needed (but not mentioned here) and most frequently, non-parametric tests are used for gene expression analysis. Given the lack of raw data, I am not able to judge the scientific quality of these results. Looking at the standard deviation I would think that the assumptions for parametric statistics are not met.
How come the controls of week 4 and week 6 have exactly the same relative expression value and exactly a variance of zero? This cannot be possible if the analysis is performed correct. Of course the authors can set the control group at 1, but in that case, multiple controls should at least result in a measurable standard deviation similar in size compared to the other groups. If the wk4 data are relative to the wk4 control and the wk6 data to the wk6 control, then the authors cannot perform statistical tests between the relative values of wk4 and wk6, as it could be the control, rather than the treatment which has changed.
Minor comments:
For some instruments e.g. the weight scale, the microscope, only the supplier is given. The type of instrument should also be mentioned.
Author Response
Response to Reviewer 1 Comments
Thank you for the valuable inputs and comments that will permit to improve the quality of this paper. We have corrected all grammars and other editing as per reviewer’s comments.
Point 1: In the present paper, the authors investigated the effect of corticosterone (CORT) on telomere length and meat quality traits in chickens. This article may be of relevance to the field, providing alternative tests for meat quality. However, the authors should improve the statistical analysis, and should provide more insight into the data. In addition, the gene expression analysis is not state of the art and the results may therefore be biased due to the method of normalization used (single reference gene only).
Response 1: The statistical analysis has been improved following the series of observations in your subsequent comments. The gene expressions for COX6A1, UCP3, SAAL1 and CRP had been reanalyzed using two reference genes (GAPDH and β-actin). However, for quantification of the telomere length, only one reference gene was used. This is in accordance with the description of Herborn et al. (2018).
Point 2: Although it is not entirely clear from the introduction. I assume that the authors used the CORT diet as a proxy for animal stress, to investigate biomarkers for meat quality traits. If this is correct, the authors should state this more clearly in the introduction. In addition, if this is the case, the authors should indicate whether this CORT diet is equivalent to the stressors that will act on chickens in a real life situation. The authors report some effects of the CORT diet on Telomere length, suggesting telomere length as a biomarker, but it should be discussed whether this biomarker will be sensitive enough for real life stressors.
Response 2: It has been stated more clearly in the introduction that CORT was used as a proxy for animal stress to investigate biomarkers of meat quality traits and gastrointestinal tract physiology. The CORT diet mimics real-life stressors and telomere length as a biomarker of stress was discussed to be sensitive enough for the real-life stressors.
Point 3: The treatment group received CORT that was diluted in 20ml of ethanol and mixed in the feed. Did the control group also receive 20ml of ethanol mixed in the feed? This may be essential, as ethanol may also have an effect on the liver and may thus bias the results of the treatment group.
Response 3: The control group diet was mixed with 20 ml of ethanol. Therefore, the control group also received 20ml of ethanol mixed in their feed.
Point 4: I have some comments on the statistical design of the experiments. The authors used t-tests and checked the assumption of normality, but they do not mention how this was checked and neither do they report whether the residuals were also normally distributed and whether the variation was homoscedastic. Given the relative small sample size, I doubt that the assumptions will be met for all tests. As such, I would strongly suggest the authors to disclose all raw data and the results of the statistical tests in a supplemental file.
Response 4: The assessment of histogram distribution and Quantile-Quantile (Q-Q) plots for model residual was used for the assumption of normality. The raw data for this experiment and the results of the statistical shall be made available.
Point 5: In addition, the authors should provide the exact p-values, instead of merely stating p<0.05, at least for all –values under between 0.001 and 0.05. Furthermore, the authors should perform multiple hypothesis correction. Given the evaluation of a large number of hypothesis on a relatively small number of samples, the risk of false positives is high.
Response 6: The exact p-values had been accordingly provided. The multiple hypothesis correction was conducted for all the features using Benjamini-Hochberg (BH) correction method (P < k/m x α). Doing this, we found out that rejection and acceptance of the null hypothesis on all the variables as presented in our findings were true.
Point 7: The authors should not mention the standard error of the mean (SEM). Reporting SEM is considered as outdated as it disguises the real variation that is present in the samples. The actual standard deviation for each of the measures or the 95% confidence intervals, if non-parametric statistics would be considered should be provided.
Response 7: We have removed the standard error of the mean (SEM) earlier used in this study and used standard deviations for each of the measures in the tables.
Point 8: I would also urge to make the data more visible. Tables with only the means harbour very little room for interpretation. A better representation would be given using box-plots with the actual data points plotted on top of them. That would give the readers a good idea of the real distribution and the effect of each of the tested hypotheses.
Response 8: The data is now made visible with the use of standard deviation. We actually used 2 x 2 factorial design for this experiment and most results presented in the tables were from 2 x 2 factorial analysis not t-test. T-test was only used for the gene expression. We therefore presented these results using table format with their respective standard deviations (±SD).
Point 9: The telomere lengths seem to be higher in the week 6 versus week 4 results. This is counterintuitive, but could of course be due to variation in the measurement procedure, but this is not clear and not discussed. Giving the raw data and a visual representation of the data (boxplots) would enhance the understanding of this variation.
Response 9: The lower telomere length at week 4 than week 6 was expected. This could be as a result of interaction between age and early metabolic activities. Effect of metabolic activities that are usually higher at early age could be responsible. We have discussed similar report in our previous study (Badmus et al., 2021). The present form of presentation could be appropriate.
Point 10: The authors should carefully check the MIQE guidelines (https://academic.oup.com/clinchem/article/55/4/611/5631762) for reporting the experimental procedures and data of the quantitative PCR experiments. These guidelines provide a checklist of the minimal information that should be given when using these methods.
Response 10: We carefully checked MIQE guidelines for reporting experimental procedures and data of the quantitative PCR experiments and found it helpful. For telomere length, we used quantitative PCR and reverse transcriptase PCR for the gene expression.
Point 11: I have serious doubts about the gene expression analysis results. The authors only used a single reference gene (GAPDH) to normalize their data. This is a very outdated method of performing RT-qPCR and may result in biased results as the authors and by no means assume that the reference gene itself is not subject to variation between the experimental procedures. Already since 2002, the RT-qPCR field urges researchers to pre-validate reference gene stability from an initial set of at least 6-10 reference genes and use at least the two most stable reference genes for normalization.
Response 11: Two reference genes (GAPDH and β-actin) had now been employed after testing of some reference genes. These two reference genes were employed following the testing of several reference genes including 36B4 and found out that the current GAPDH and β-actin were most stable in our laboratory. The average of the two reference genes was taken using geometric mean and then used to normalize the test genes.
Point 12: The graphs of the RT-qPCR results should be shown as boxplots instead of barcharts, as the latter only hide relevant information.
Response 12: We have employed boxplots instead of bar charts for the gene expression.
Point 13: I am a bit surprised that standard t-tests could be used on the RT-qPCR data. These data are frequently logarithmical in nature, hence transformation is often needed (but not mentioned here) and most frequently, non-parametric tests are used for gene expression analysis. Given the lack of raw data, I am not able to judge the scientific quality of these results. Looking at the standard deviation I would think that the assumptions for parametric statistics are not met.
Response 13: The gene expression is now presented using Box-plot.
Point 14: How come the controls of week 4 and week 6 have exactly the same relative expression value and exactly a variance of zero? This cannot be possible if the analysis is performed correct. Of course the authors can set the control group at 1, but in that case, multiple controls should at least result in a measurable standard deviation similar in size compared to the other groups. If the wk4 data are relative to the wk4 control and the wk6 data to the wk6 control, then the authors cannot perform statistical tests between the relative values of wk4 and wk6, as it could be the control, rather than the treatment which has changed.
Response 14: We previously set the control group at 1 and this was used to run t-test. However, now we have used box-plot as you suggested.
Minor comments:
Point 15: For some instruments e.g. the weight scale, the microscope, only the supplier is given. The type of instrument should also be mentioned.
Response 15: We have provided the type of the instrument for the weight scale and microscope used in the paper.
Reviewer 2 Report
This study revealed the telomere length, apoptotic and inflammatory genes as potential biomarkers of gastrointestinal tract pathology and meat quality traits in chronic stress in chickens. The research topic is innovative and meaningful; results are significant; data analysis is correct; and English writing is proficiency. Thus, this article is advised to be acceptable but need revisions.
Detailed comments are as follows:
- Check the manuscript carefully and correct grammar mistakes, punctuation, diction and typos.
Line 25: “Chickens were fed with diet…”, “diet” should be corrected as “a diet” or “diets”.
Line 28: P < 0.05, “P” should be italic. Same in other places.
Line 133: CIORT
Line 229: A part of…
Line 235: 60 ℃. Same in lines 278-280.
Line 464: bodyweight
Line 498: could be attributed to increased…
- How did the authors determine the feeding period of CORT? Why did it start on day 15 of age and last 4 weeks? Also how did the authors determine the concentration of CORT?
- Please simplified the Materials and Methods section. Some description was too detailed.
- Table 3: Please center the value “214.74b” in the table.
- Table 7: It is necessary to also display the H &E staining images of liver histopathology, not just the statistical data.
- Which statistical method of correlation analysis was used in Table 10? Please present it in the table note.
- Figures 1-4 are suggested to be redrew, as they are low pixeled, particularly the axis titles.
- The conclusion section should be simplified and highlighted the key findings.
Author Response
Response to Reviewer 2 Comments
Thank you for the valuable inputs and comments that will permit to improve the quality of this paper. We have corrected all grammars and other editing as per reviewer’s comments.
Point 1: Check the manuscript carefully and correct grammar mistakes, punctuation, diction and typos.
Line 25: “Chickens were fed with diet…”, “diet” should be corrected as “a diet” or “diets”.
Line 28: P < 0.05, “P” should be italic. Same in other places.
Line 133: CIORT
Line 229: A part of…
Line 235: 60 ℃. Same in lines 278-280.
Line 464: bodyweight
Line 498: could be attributed to increased…
Response 1: All the grammatical mistakes, punctuations, diction and typographical errors had been properly checked and corrected as highlighted above.
Point 2: How did the authors determine the feeding period of CORT? Why did it start on day 15 of age and last 4 weeks? Also how did the authors determine the concentration of CORT?
Response 2: The supplemental level of CORT (30mg/kg diet) according Hu et al., 2010 was employed. The concentration of CORT (30mg/kg diet) which was noted to trigger changes in their broiler chicken performances was adopted.
Point 3: Please simplified the Materials and Methods section. Some description was too detailed.
Response 3: Some parts of the Materials and Methods section had been simplified.
Point 4: Table 3: Please center the value “214.74b” in the table.
Response 4: The value “214.74b” in the table 3 had been properly centred.
Point 5: Table 7: It is necessary to also display the H &E staining images of liver histopathology, not just the statistical data.
Response 5: The image displaying the H & E staining of liver histopathology had been displayed in Figure 1.
Point 6: Which statistical method of correlation analysis was used in Table 10? Please present it in the table note.
Response 6: We used Pearson coefficient for the correlation analysis and it has been presented in the table 10 footnote.
Point 7: Figures 1-4 are suggested to be redrew, as they are low pixeled, particularly the axis titles.
Response 7: Figure 1-4 had been redrawn using box plot.
Point 8: The conclusion section should be simplified and highlighted the key findings.
Response 8: The conclusion section has been simplified with the highlight of the key findings.
Round 2
Reviewer 1 Report
The authors have adressed my comments sufficiently. They promised to submit the raw data as supplement, but they have not submitted this as part of their revision.
Author Response
Point 1. The authors have adressed my comments sufficiently. They promised to submit the raw data as supplement, but they have not submitted this as part of their revision.
Response 1: We appreciate your comment. Kindly find attached, the raw data as supplementary data as part of our revision for our paper. We hope it could've of help.
Thank you for your efforts on our paper and have a wonderful time.

Round 3
Reviewer 1 Report
I do not have further comments